# Age and Sex Specific Prevalence of Clinical and Screen-Detected Atrial Fibrillation in Hospitalized Patients

**DOI:** 10.3390/jcm10214871

**Published:** 2021-10-22

**Authors:** Laurent Roten, Eleni Goulouti, Anna Lam, Elena Elchinova, Nikolas Nozica, Alessandro Spirito, Severin Wittmer, Mattia Branca, Helge Servatius, Fabian Noti, Jens Seiler, Samuel H Baldinger, Andreas Haeberlin, Stefano de Marchi, Babken Asatryan, Nicolas Rodondi, Jacques Donzé, Drahomir Aujesky, Hildegard Tanner, Tobias Reichlin, Peter Jüni

**Affiliations:** 1Department of Cardiology, Inselspital, Bern University Hospital, University of Bern, 3010 Bern, Switzerland; eleni.goulouti@insel.ch (E.G.); annalam@gmx.ch (A.L.); elena.elchinova@insel.ch (E.E.); nikolas.nozica@insel.ch (N.N.); alessandro.spirito@insel.ch (A.S.); sevi.wittmer@gmail.com (S.W.); helge.servatius@insel.ch (H.S.); fabian.noti@insel.ch (F.N.); jens.seiler@insel.ch (J.S.); samuel.baldinger@insel.ch (S.H.B.); andreas.haeberlin@insel.ch (A.H.); stefano.demarchi@insel.ch (S.d.M.); babken.astryan@insel.ch (B.A.); hildegard.tanner@insel.ch (H.T.); tobias.reichlin@insel.ch (T.R.); 2Clinical Trials Unit, University of Bern, 3012 Bern, Switzerland; mattia.branca@ctu.unibe.ch; 3Sitem Center for Translational Medicine and Biomedical Entrepreneurship, University of Bern, 3010 Bern, Switzerland; 4Department of General Internal Medicine, Inselspital, Bern University Hospital, University of Bern, 3010 Bern, Switzerland; nicolas.rodondi@insel.ch (N.R.); DrahomirAntonin.Aujesky@insel.ch (D.A.); 5Institute of Primary Health Care (BIHAM), University of Bern, 3012 Bern, Switzerland; 6Department of Medicine, Neuchâtel Hospital Network, 2000 Neuchâtel, Switzerland; jacques.donze@rhne.ch; 7Department of Internal Medicine, Brigham and Women’s Hospital, Harvard Medical School, Boston, MA 02115, USA; 8Applied Health Research Centre (AHRC), Li Ka Shing Knowledge Institute of St. Michael’s Hospital, Department of Medicine, University of Toronto, Toronto, ON M5S, Canada; peter.juni@utoronto.ca

**Keywords:** atrial fibrillation, screening, paroxysmal, subclinical, silent, Holter ECG, prospective, cohort, hospitalized, age, sex

## Abstract

Background: The prevalence of atrial fibrillation (AF) is high in older patients. The present study aimed to estimate the age and sex specific prevalence of clinical and screen-detected atrial fibrillation (AF) in hospitalized patients. Methods: The STAR-FIB cohort study was a prospective cohort study recruiting participants from a large source population of hospitalized patients aged 65–84 years. The estimated size of the source population was 26,035 (95% CI 25,918–26,152), and 795 consenting patients without clinical AF were included in the cohort study after stratification by sex and age (49.2% females; mean age 74.7 years). Patients in the cohort study underwent three seven-day Holter ECGs in intervals of two months to screen for AF. Results: In the source population, the estimated prevalence of clinical AF was 22.2% (95% CI 18.4–26.1), 23.8% for males (95% CI 20.9–26.6) and 19.8% for females (95% CI 17.3–22.4; *p* for difference between sexes, 0.004). There was a linear trend for an increase in the prevalence of clinical AF with increasing age, overall and in both sexes. In the cohort study, AF was newly diagnosed in 38 patients, for an estimated prevalence of screen-detected AF of 4.9% overall (95% CI 3.3–6.6), 5.5% in males (95% CI 3.2–7.8) and 4.0% in females (95% CI 2.0–6.0; *p* for difference between sexes, 0.041). The estimated prevalence of screen-detected AF in the source population was 3.8% overall, 4.2% in males and 3.2% in females. Conclusion: In a large hospital-based patient population aged 65–84 years, the prevalence of clinical AF and of screen-detected AF was 22.2% and 3.8%, respectively, and significantly higher in males than females.

## 1. Introduction

Ischemic stroke is a frequent first manifestation of atrial fibrillation and may be prevented by timely initiation of anticoagulation therapy. Patients with asymptomatic atrial fibrillation have a worse outcome than patients with symptomatic atrial fibrillation and the economic burden of undiagnosed atrial fibrillation is high [1,2,3]. European guidelines state that systematic screening for atrial fibrillation should be considered in individuals aged ≥ 75 years, or those at high risk of stroke (Class IIA recommendation; level of evidence B) [4]. Patients with screen-detected atrial fibrillation and a single-lead ECG of ≥30 s or a 12-lead ECG showing atrial fibrillation should be anticoagulated if they have an increased thromboembolic risk according to their CHA_2_DS_2_VASc score.

Typically, screening studies for atrial fibrillation have been performed in large communities [5], primary care [6], outpatient clinics [7], pharmacies [8], or by consumer volunteers [9]. Hospitalized patient populations are generally exempted from screening studies, although they have a particularly high prevalence of clinical, i.e., known atrial fibrillation and a worse risk profile [10]. The prevalence of screen-detected atrial fibrillation should be interpreted in relation to the prevalence of clinical atrial fibrillation in the screened population. Due to selection bias, most screening studies are unable to adequately describe the prevalence of clinical atrial fibrillation in the source population [5]. The prevalence of clinical atrial fibrillation increases with age, and several studies have reported an association between age and the prevalence of screen-detected atrial fibrillation. However, no study has investigated in detail the interplay of age with the prevalence of clinical and screen-detected atrial fibrillation in hospitalized patients.

## 2. Methods

The STAR-FIB cohort study is a hospital-based, prospective cohort study and part of the STAR-FIB study program. The details of the study program have been described elsewhere, including a detailed description of the recruitment process [11].

In brief, a source population of consecutive patients aged 65–84 years admitted to the Departments of General Internal Medicine, Cardiology and Ophthalmology at our tertiary care hospital was evaluated for the presence of clinical atrial fibrillation during the study recruitment period from 19 January 2015 to 26 June 2019. In case of multiple admissions, only the first hospital admission per patient during the study period was considered. Data on age, sex and the presence of clinical atrial fibrillation and type of atrial fibrillation of the source population according to medical records were recorded in a recruitment log for all patients aged 65–84 years admitted during active recruitment periods (*N* = 11,470). Patients without clinical atrial fibrillation in this source population were screened for inclusion in the prospective cohort study, with the intention to recruit 100 males and 100 females in each of four age bands (≥65 to <70, ≥70 to <75, ≥75 to <80, and ≥80 to <85 years). Recruitment of male participants aged < 80 years was faster than recruitment of the remaining subgroups. We therefore randomly selected calendar weeks during which males aged < 80 years would be recruited, whereas the remaining subgroups were continuously recruited. Recruitment was capped once the necessary number of participants was reached in a subgroup, and recruitment was considered closed for this subgroup. Active recruitment periods were defined as periods during which all patients of an age and sex specific subgroup were systematically screened for participation in the prospective cohort study at the time of hospital admission. Non-active recruitment periods were defined as periods during which all patients of an age and sex specific subgroup were not screened for the reasons described above.

Exclusion criteria for the prospective cohort study, apart from clinical atrial fibrillation, were an indication for long-term anticoagulation therapy, recent (<3 months) acute coronary syndrome or hospitalization due to heart failure, and recent or planned percutaneous coronary revascularization, transcutaneous aortic valve replacement, and major cardiac or non-cardiac surgery.

All participants included in the prospective cohort study underwent three seven-day Holter ECGs to screen for atrial fibrillation in intervals of two months. The first seven-day Holter ECG was recorded upon hospital discharge. If atrial fibrillation was diagnosed, the subsequent seven-day Holter ECGs were cancelled. The primary endpoint was a diagnosis of atrial fibrillation or atrial flutter of more than 30 s duration in a seven-day Holter ECG, on a rhythm strip or on any conventional 12-lead ECG. The study complies with the Declaration of Helsinki and was approved by the locally appointed ethics committee of the Canton of Bern (KEK-BE 257/14). All study participants provided written informed consent. 

For the purpose of this analysis, we use the term screen-detected atrial fibrillation for any new diagnosis of atrial fibrillation after study inclusion, irrespective of the modality of diagnosis. The term clinical atrial fibrillation is used to describe the prevalence of known atrial fibrillation in the source population.

## 3. Statistical Analyses

A more detailed description of the statistical analysis can be found elsewhere [11]. The design allowed for the estimation of the number of inpatients aged 65–84 years admitted during the study period from 19 January 2015 to 26 June 2019, which was the estimated source population of the prospective cohort study. Using data from the recruitment log, we then estimated the age and sex specifics, and the overall prevalence of clinical atrial fibrillation and of paroxysmal atrial fibrillation in the source population with 95% confidence intervals (CI). For each age and sex subgroup, we obtained the variances of prevalence estimates as the sum of the relevant variances of estimates used to derive the prevalence [12]. We used fixed-effect linear regression with the inverse of the variance as analytical weights to derive p-values for difference in prevalence between males and females, and p-values for trends across age subgroups.

We estimated the age and sex-specific prevalence of screen-detected atrial fibrillation and corresponding 95% CIs were based on the number of participants with a new diagnosis of atrial fibrillation. To derive the age and sex-specific prevalence, and the overall prevalence of screen-detected atrial fibrillation, we used the svy family of commands in Stata that took the complex recruitment strategy into account based on appropriate analytical weights [13]. The use of analytical weights resulted in independence of estimates from the uniform age and sex distribution related to the recruitment strategy. Prevalence estimates therefore reflect the true age and sex distribution of potentially eligible inpatients hospitalized during the entire recruitment period. To estimate the prevalence of screen-detected atrial fibrillation in the source population, we divided the age and sex specific prevalence estimates of screen-detected atrial fibrillation in the examined population by the age and sex specific proportion of patients without clinical atrial fibrillation in the source population. If specific estimates were combined to derive overall prevalence estimates (for example, combining age-specific estimates to derive an overall estimate across all subgroups), the standard error to derive confidence intervals of the overall prevalence estimate was based on the square root of the sum of the variances of individual estimates. All statistical analyses were performed using Stata 16.1 (StataCorp, College Station, TX, USA).

## 4. Results

### 4.1. Prevalence Estimates of Clinical Atrial Fibrillation in the Source Population

The estimated number of patients aged 65–84 years admitted to the hospital during the study recruitment period was 26,035 (95% CI 25,918 to 26,152). The estimated prevalence of clinical atrial fibrillation in the source population was 22.2% overall (95% CI 18.4 to 26.1), and 23.8% for males (95% CI 20.9 to 26.6) versus 19.8% for females (95% CI 17.3 to 22.4; *p* for difference between sexes, 0.004; Figure 1). There was a linear trend for an increase in the prevalence of clinical atrial fibrillation with increasing age, overall and in both sexes (*p* for trend ≤ 0.024).

The prevalence of paroxysmal atrial fibrillation in the source population was 10.4% overall (95% CI 6.9 to 13.9), without a significant difference between sexes. Again, there was a linear trend for an increase in the prevalence of paroxysmal atrial fibrillation with increasing age, overall and in both sexes (*p* for trend ≤ 0.046; Figure 1). The percentage of paroxysmal atrial fibrillation among patients with any type of clinical atrial fibrillation was 46.7% overall (95% CI 38.1 to 55.3), and 43.3% for males (95% CI 37.1 to 49.4) versus 53.1% for females (95% CI 47.1 to 59.1; *p* for difference between sexes, 0.002). Overall, and in the subgroup of males, this rate was not associated significantly with age, but there was a linear trend towards a decrease with increasing age in females, which was of borderline significance (*p* for trend = 0.041; Figure 1).

### 4.2. Characteristics of Patients Included in the Cohort Study

The patient characteristics are shown in Table 1. We recruited 91 females aged ≥ 80, and 100 females each in the other three age groups. We recruited 102 males each in the age groups ≥ 70 to <75 and ≥75 to <80 years, and 100 males each in the remaining age groups. Therefore, 795 patients were included in the cohort study, 391 females (49.2%) and 404 males. The mean age of the entire cohort was 74.7 ± 5.6 years.

### 4.3. 7-Day Holter ECGs in Patients Included in the Cohort Study

We performed a total of 2,077 seven-day Holter ECGs: three in 616 patients (77.5%); two in 51 patients (6.4%); and one in 127 patients (16.0%). One patient had no seven-day Holter ECG because of a diagnosis of atrial fibrillation after study inclusion but before the first seven-day Holter ECG. Figure 2 shows the reasons for not completing three seven-day Holter ECGs. The median time interval between the first and second seven-day Holter ECG was 94 days (interquartile range 68; 117), and 64 days (61; 72) between the second and third. The mean cumulative duration of an analyzable ECG signal recording in seven-day Holter ECGs was 414 ± 136 h per patient. In the first, second and third seven-day Holter ECG, analyzable ECG signals were recorded for 156 ± 28 h, 160 ± 19 h, and 161 ± 18 h, respectively.

### 4.4. Screen-Detected Atrial Fibrillation

We diagnosed 38 new cases of atrial fibrillation (Appendix A). Of these, 29 cases were detected with a seven-day Holter ECG (76.3%), 15 in the first, five in the second, and nine in the third seven-day Holter ECG. Nine cases were detected in the absence of a seven-day Holter ECG as described in Appendix A.

In screen-positive participants, the cumulative median duration of analyzable ECG signal recordings until diagnosis of atrial fibrillation was 163 h (IQR 29 to 330). The median duration to diagnosis of atrial fibrillation in first, second and third seven-day Holter ECGs was 32 h (4 to 68), 97 h (25 to 134), and 14 h (2 to 128), respectively (*p* = 0.562; Appendix A). The median number of atrial fibrillation episodes in screen-positive seven-day Holter ECGs was 2 (1 to 6), while the median cumulative duration of atrial fibrillation was 198 min (41 to 544). Appendix A presents the number of episodes and cumulative duration of atrial fibrillation separately for the first, second, and third seven-day Holter ECG.

### 4.5. Prevalence of Screen-Detected Atrial Fibrillation

The estimated prevalence of screen-detected atrial fibrillation in the prospective cohort study was 4.9% overall (95% CI 3.3 to 6.6), and was significantly higher for males (5.5%; 95% CI 3.2 to 7.8) than for females (4.0%; 95% CI 2.0 to 6.0; *p* for difference between sexes, 0.041; Figure 3 and Appendix A). There was a linear trend of borderline significance for an increase in prevalence associated with age in males (*p* for trend = 0.052), but not in females (Figure 3). The estimated prevalence of screen-detected atrial fibrillation in the source population was 3.8%, 4.2% for males and 3.2% for females (Graphical abstract).

## 5. Discussion

This study shows a high prevalence of clinical atrial fibrillation of 22.2% in a hospitalized patient population aged 65–84 years. Studies attempting to define the prevalence of clinical atrial fibrillation have reported a prevalence of 1–4% in the general population aged 65 years and 6–15% in those aged 80 years [14,15]. The corresponding prevalence of 15% and 25%, respectively, in our population of hospitalized patients is notably higher and a reflection of the sicker patient population included. Similar to other studies in non-hospitalized patients, we found a higher prevalence of clinical atrial fibrillation in males, and a typical age-dependent rise in both sexes [15]. but with higher prevalence in all age groups in hospitalized patients as compared with the community.

The estimated prevalence of screen-detected atrial fibrillation in our cohort was 4.9%, and significantly higher in males compared to females. There was also a statistical trend towards an age-dependent increase in males, which was not observed in females. These differences between sexes and age groups have not been reported previously. The prevalence of screen-detected atrial fibrillation in our cohort is within the range of other studies using similar screening methods. The mSToPS trial enrolled outpatients with a median age of 73.5 years and a median CHA_2_DS_2_-VASc score of 3 [16]. Patients wore an ECG patch in the active arm for two weeks twice, corresponding to a total of four weeks of continuous ECG monitoring. With this approach, the prevalence of screen-detected atrial fibrillation was 5.1%, and almost identical to our population. In the ARIC study, the same ECG monitoring device was used once in a population with a mean age of 79 years and a mean CHA_2_DS_2_-VASc score of 3.8. This study reported a lower prevalence of screen-detected atrial fibrillation of 2.5% [16]. The STROKESTOP study included only patients aged 75 years with a mean CHA_2_DS_2_-VASc score of 3.4 [5]. Using an initial 12-lead ECG and twice daily 30-s one-lead ECGs for 14 days, the study found a prevalence of screen-detected atrial fibrillation of 3.0%. The REHEARSE-AF study used 30-s ECG recordings twice weekly and upon symptoms for 12 months [6]. This study reported a prevalence of screen-detected atrial fibrillation of 3.8% in patients with a mean age of 72.6 years and a mean CHA_2_DS_2_-VASc score of 3.0.

All aforementioned studies recruited patients from general practitioner records [6], general communities [5], or health insurance plans [16]. We are aware of only one study in hospitalized patients which assessed the feasibility of screening a geriatric population for atrial fibrillation with a hand-held device and reported a prevalence of screen-detected atrial fibrillation of 13% [17]. The prevalence of clinical atrial fibrillation in the populations of the above-mentioned studies is markedly lower than in our study: 14.2% in the ARIC study and 9.3% in the STROKESTOP study [5,16]. However, most screening studies do not report the prevalence of clinical atrial fibrillation in their source population. A strength of our study is the detailed information on the prevalence of clinical atrial fibrillation in the source population for both sexes and different age groups. To understand the full burden of atrial fibrillation, the prevalence of clinical and screen-detected atrial fibrillation have to be summed up.

Screening for silent atrial fibrillation by repeat seven-day Holter ECGs underestimates the prevalence of this condition. A comprehensive evaluation of rhythm monitoring strategies for atrial fibrillation screening using 12 months of follow-up data from implantable cardiac monitors estimated the sensitivity of three repeat seven-day-Holter ECGs to be about 33% [18]. Considering this data and the fact that only about 80% of the patients included in our prospective cohort study completed three seven-day-Holter ECGs, the true prevalence of silent atrial fibrillation in our cohort would likely be approximately three times higher with continuous monitoring for one year and amount to 12–15%. In patients with cryptogenic, ischemic stroke at a mean age of 61.6 years, the cumulative incidence of silent atrial fibrillation diagnosed with an implantable cardiac monitor was 12.4% at 12 months [19]. Other studies with implantable cardiac monitors in older patient populations aged 71–76 years and with additional risk factors found the prevalence of silent atrial fibrillation to be 20–30% at 12 months [20,21,22,23]. These studies targeted patients at particularly high risk of atrial fibrillation, reflected by the high mean CHA_2_DS_2_-VASc score of 3.9 to 4.6 in these studies.

Recently, the results of two randomized controlled trials have been published, which investigated the benefit of screening for atrial fibrillation with subsequent anticoagulation of screen-positive participants. In the STROKESTOP study mentioned above, screening was only performed at study inclusion for two weeks and was positive in 3% of screened patients [24]. Due to the study design, only half of the participants in the active arm were actually screened for atrial fibrillation. Nevertheless, screening showed a small net benefit compared to standard of care after several years of follow-up. The second trial was the LOOP study, in which an implantable cardiac monitor resulted in screen-positive atrial fibrillation in almost one third of participants in the active arm [23]. The primary endpoint of time to first stroke or systemic embolism failed to show a benefit. Importantly, the incidence of atrial fibrillation was 12% in the comparator arm of usual care, and only 6% of patients in the active arm had maximal duration of atrial fibrillation episodes of >24 h at any time [25]. In fact, short episodes of atrial fibrillation during continuous monitoring as recorded by an implantable cardiac monitor seem to confer a lower risk compared to longer episodes or higher atrial fibrillation burden [26]. 

Although implantable cardiac monitors may be the gold standard for atrial fibrillation screening, additional ECG evidence is required according to the latest European guidelines, before recommending initiation of anticoagulation therapy, particularly if the burden of atrial fibrillation is low. Repeat seven-day Holter ECG is an inexpensive and readily available, non-invasive screening method. If atrial fibrillation is detected by repeat 7-day Holter ECG as in our study, these patients usually have an atrial fibrillation burden that is high enough to justify oral anticoagulation therapy [18]. A Holter ECG has the additional advantage to provide unequivocal ECG documentation of atrial fibrillation as required by the guidelines [4]. Accordingly, anticoagulation therapy was recommended to all patients with screen-detected atrial fibrillation in our study cohort.

This study has several limitations. First, it is a single center study. Second, the aim of this study was to determine the prevalence of silent atrial fibrillation in hospitalized patients, and our results should not be generalized to outpatients or the general community. Third, selection bias, potentially resulting in an overestimation of the prevalence of silent atrial fibrillation, cannot be excluded. Fourth, the implantation of a cardiac monitor would be the gold standard for atrial fibrillation screening, as it would result in higher detection rates.

In conclusion, the prevalence of clinical atrial fibrillation in hospitalized patients aged 65–84 years was 22.2% and significantly higher in males than females. It showed the typical age-dependent rise in both sexes. The estimated prevalence of screen-detected atrial fibrillation in our source population of hospitalized patients aged 65–84 years was 3.8%, and was higher in males than in females. Systematic screening for atrial fibrillation in elderly hospitalized patients followed by appropriate therapy could be promising in reducing the morbidity associated with atrial fibrillation in the population.

## Figures and Tables

**Figure 1 jcm-10-04871-f001:**
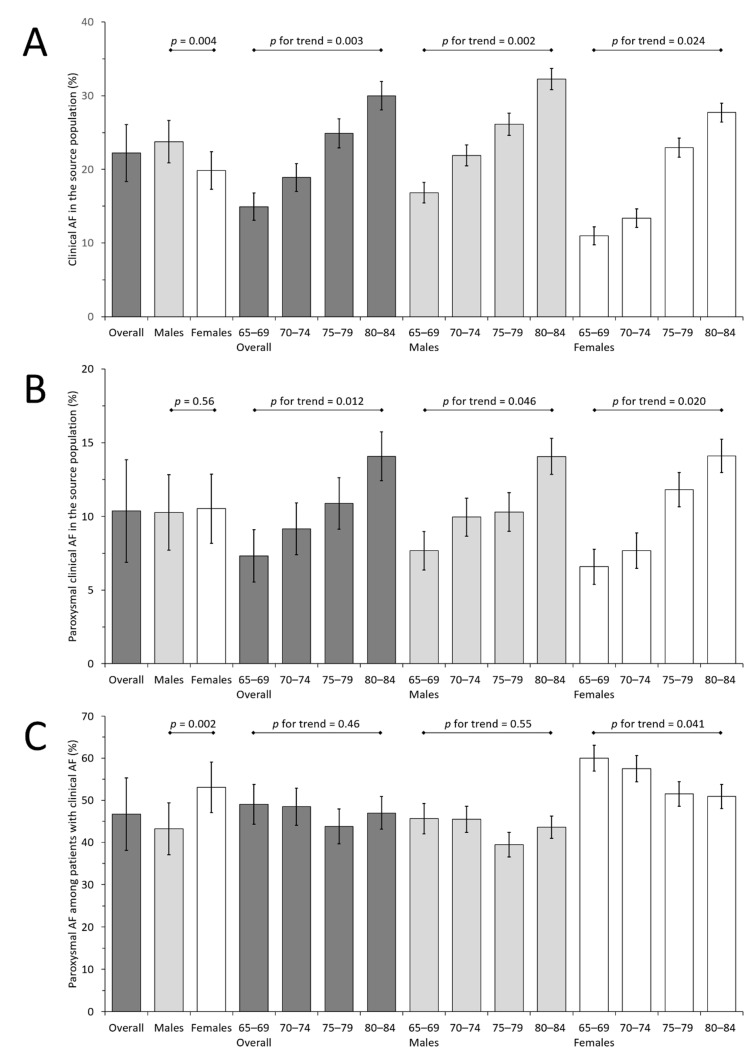
Prevalence estimates in percentages with 95% confidence intervals of clinical atrial fibrillation (including both paroxysmal and non-paroxysmal atrial fibrillation) in the source population (**A**), of paroxysmal clinical atrial fibrillation in the source population (**B**), and of paroxysmal atrial fibrillation among patients with clinical atrial fibrillation (**C**). Prevalence estimates are shown overall, and by sex and age subgroups. 95% confidence intervals of prevalence estimates for the overall population and for males and females combined are wider than 95% confidence intervals for individual age/sex subgroups, as the complex recruitment strategy was taken into account when calculating variances (see Section 2). AF: atrial fibrillation.

**Figure 2 jcm-10-04871-f002:**
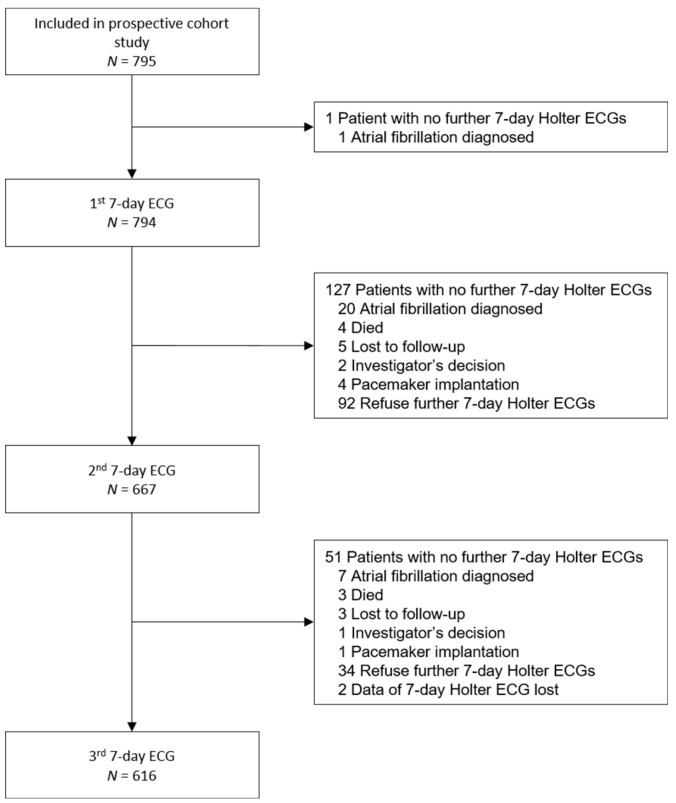
Flow chart showing the number of 7-day Holter ECGs performed in the cohort study and reasons for not completing three 7-day Holter ECGs.

**Figure 3 jcm-10-04871-f003:**
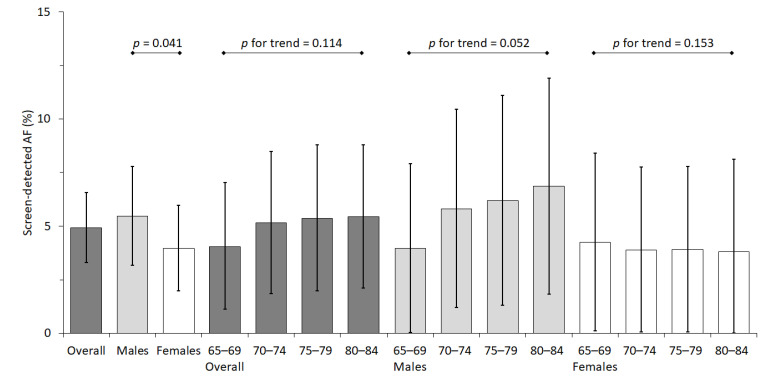
Prevalence estimates of screen-detected atrial fibrillation as percentages with 95% confidence intervals in the prospective cohort study. AF: atrial fibrillation.

**Table 1 jcm-10-04871-t001:** Patient characteristics.

	Cohort *N* = 795	No AF *N* = 757	AF *N* = 38	*p* Value
Age, years	74.7 ± 5.6	74.7 ± 5.6	74.9 ± 5.5	0.859
Sex, female	391 (49%)	376 (50%)	15 (39%)	0.247
Arterial hypertension	557 (70%)	531 (70%)	26 (68%)	0.856
Diabetes mellitus	152 (19%)	147 (19%)	5 (13%)	0.404
Dyslipidemia	419 (53%)	398 (53%)	21 (55%)	0.868
Coronary artery disease	246 (31%)	233 (31%)	13 (34%)	0.719
Peripheral artery disease	53 (7%)	49 (6%)	4 (11%)	0.311
Congestive heart failure	18 (2%)	18 (2%)	-	1.0
Previous thrombotic event	121 (15%)	116 (15%)	5 (13%)	0.821
Stroke	65 (8%)	60 (8%)	5 (13%)	0.228
Transient ischemic attack	53 (7%)	53 (7%)	-	0.169
Peripheral embolism	9 (1%)	9 (1%)	-	1.0
Palpitations within last 12 months	187 (24%)	176 (23%)	11 (29%)	0.434
Body mass index, kg/m^2^	26.5 ± 4.6	26.4 ± 4.5	28.2 ± 6.7	0.03
LV ejection fraction, %	62.1 ± 6.0	62.1 ± 5.9	60.8 ± 7.3	0.226
CHA_2_DS_2_Vasc Score	3.5 ± 1.4	3.5 ± 1.4	3.4 ± 1.4	0.792
1	-	-	-	
2	47 (6%)	44 (6%)	3 (8%)	
3	171 (22%)	163 (22%)	8 (21%)	
4	216 (27%)	206 (27%)	10 (26%)	
5	192 (24%)	184 (24%)	8 (21%)	
6	99 (12%)	93 (12%)	6 (16%)	
7	49 (6%)	46 (6%)	3 (8%)	
8	17 (2%)	17 (2%)	-	
9	3 (0%)	3 (0%)	-	
Drug therapy				
Anticoagulation therapy	7 (1%)	7 (1%)	-	0.382
Acetylsalicylic acid	402 (51%)	385 (51%)	17 (45%)	0.508
Betablocker	285 (36%)	271 (36%)	14 (37%)	0.865
Calcium channel blocker	152 (19%)	144 (19%)	8 (21%)	0.678
Digoxin	1 (0%)	1 (0%)	-	1.0
ACE inhibitor	184 (23%)	178 (24%)	6 (16%)	0.328
Angiotensin receptor blocker	231 (29%)	218 (29%)	13 (34%)	0.468
Aldosterone antagonist	24 (3%)	24 (3%)	-	0.623
Statin	409 (51%)	387 (51%)	22 (58%)	0.506
Diuretics	195 (25%)	184 (24%)	11 (29%)	0.562

AF: atrial fibrillation; LV: left ventricular.

## Data Availability

The data presented in this study are available on request from the corresponding author.

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
