# Peer review of "Age and Sex Specific Prevalence of Clinical and Screen-Detected Atrial Fibrillation in Hospitalized Patients"

_jcm, 2021, doi:10.3390/jcm10214871_

Round 1
Reviewer 1 Report
The authors have presented their observations of the prevalence of atrial fibrillation in a large cohort hospitalized for other reasons and performed a prospective cohort study in a defined subset of those free of atrial fibrillation during hospitalization. There are several omissions in the methods in that they do not describe how atrial fibrillation was defined and identified in the hospitalized population. That information is probably in the prior STAR-FIB publications but should be summarized in the current manuscript.
A prevalence of paroxysmal atrial fibrillation in the same population was 10.4%. but it is not clear if this was a separate group or part of the overall prevalence in the source population. Again, neither the definition of paroxysmal atrial fibrillation nor how it was detected in the source population were presented, and the presentation of the data related to paroxysmal atrial fibrillation is not clear.
Perhaps the discussion could have more emphasis on the limitations of 7-day Holter monitoring to detect atrial fibrillation. This reviewer would have liked a clearer discussion of when anticoagulation therapy should be initiated based on the detection of transient atrial fibrillation.
Author Response
Reviewer 1
The authors have presented their observations of the prevalence of atrial fibrillation in a large cohort hospitalized for other reasons and performed a prospective cohort study in a defined subset of those free of atrial fibrillation during hospitalization. There are several omissions in the methods in that they do not describe how atrial fibrillation was defined and identified in the hospitalized population. That information is probably in the prior STAR-FIB publications but should be summarized in the current manuscript.
We relied on the diagnosis of atrial fibrillation according to the charts. We did not request the ECGs showing atrial fibrillation for verification, because the work load to accomplish this would have been too high (N=2’525!). Hence, the presence and type of clinical atrial fibrillation in the hospitalized cohort was identified according to patient charts, as described on line 73-76 (see below). We added … of the source population… for clarification.
Line 76-79: Data on age, sex and the presence of clinical atrial fibrillation and type of atrial fibrillation of the source population according to medical records were recorded in a recruitment log for all patients aged 65-84 years admitted during active recruitment periods (N=11’470).
A prevalence of paroxysmal atrial fibrillation in the same population was 10.4%. but it is not clear if this was a separate group or part of the overall prevalence in the source population. Again, neither the definition of paroxysmal atrial fibrillation nor how it was detected in the source population were presented, and the presentation of the data related to paroxysmal atrial fibrillation is not clear.
Data on clinical AF and on clinical, paroxysmal AF were always calculated for the source population. Figure 1 illustrates this. See also legend of Figure 1 on Lines 149-151:
Prevalence estimates in percentages with 95% confidence intervals of clinical atrial fibrillation in the source population (A), of paroxysmal clinical atrial fibrillation in the source population (B), and of paroxysmal atrial fibrillation among patients with clinical atrial fibrillation (C).
For clarification, we added “…(including both paroxysmal and non-paroxysmal atrial fibrillation)…” on line 150 and changed “…same population…” on line 157 to “…source population…”.
For further clarification, we added “…the source population…” on line 78.
Perhaps the discussion could have more emphasis on the limitations of 7-day Holter monitoring to detect atrial fibrillation. This reviewer would have liked a clearer discussion of when anticoagulation therapy should be initiated based on the detection of transient atrial fibrillation.
We think that a Holter ECG has the advantage to detect atrial fibrillation in patients with a sufficient burden to justify oral anticoagulation therapy. If instead an implantable cardiac monitor is used, atrial fibrillation is detected also in patients with only short episodes and low burden, in which the benefit of oral anticoagulation is less clear. We changed the manuscript as follows on lines 294-299:
If atrial fibrillation is detected by repeat 7-day Holter ECG as in our study, these patients usually have an atrial fibrillation burden that is high enough to justify oral anticoagulation therapy. [please include the following reference: 10.1161/CIRCULATIONAHA.119.044407] A Holter ECG has the additional advantage to provide unequivocal ECG documentation of atrial fibrillation as required by the guidelines.

Reviewer 2 Report
I would like to congratulate the authors on the results of prospective STAR-FIB cohort study. This is an interesting research paper that contributes to an important atrial fibrillation (AF) topic of clinical and screen detected AF. In general the article is well written, but there are several issues that have to be adressed:
- Line 45 – more robust data are already available (the LOOP trial), the authors have to rephrase this statement while in the meantime from the submission of their article to the review , new data were published
- Line 46 – patient with asymptomatic atrial fibrillation have worse prognosis – this is quite strong statement that has to be supported by a citation. In the first place I do not believe in asymptomatic atrial fibrillation, the main problem is that the patient are not appropriately questioned
- Line 47 – how do you recount the economic burden of undiagnosed atrial fibrillation, again we need a citation or the sentence has to start with.. we may speculate...
- Line 49 – systematic ECG screening in 75 year and older is IIaB recommendation, but opportunistic screening in ≧ 65 years old is IB. The authors may leave this statement as it is, but the level of evidence is quite weak and younger patients should be sreened and treated beforehand
- Line 214– study by Tavernier R et al. Heart 2018 Apr;104(7):588-593 should be stated, the prevalence of AF in hospitalized patients was even higher (46%)
- Add the results of the LOOP trial to the discussion section, and please comment on the treatment of screen detected AF
- Do the authors think that all screen-detected AF patients should be treated with anticoagulation? Does it depends on the lenght of the Afib episode or the risk profile of the patients?Because sceening without setting some therapeutic goal does not make sense.
Author Response
Reviewer 2
I would like to congratulate the authors on the results of prospective STAR-FIB cohort study. This is an interesting research paper that contributes to an important atrial fibrillation (AF) topic of clinical and screen detected AF. In general the article is well written, but there are several issues that have to be adressed:
- Line 45 – more robust data are already available (the LOOP trial), the authors have to rephrase this statement while in the meantime from the submission of their article to the review , new data were published
We deleted this sentence and the references in the introduction section completely and instead discuss this topic in more details in the discussion section (see below).
- Line 46 – patient with asymptomatic atrial fibrillation have worse prognosis – this is quite strong statement that has to be supported by a citation. In the first place I do not believe in asymptomatic atrial fibrillation, the main problem is that the patient are not appropriately questioned
We added the following two references:
https://doi.org/10.1016/j.ijcard.2013.07.234
https://doi.org/10.1160/th4-04-0383
- Line 47 – how do you recount the economic burden of undiagnosed atrial fibrillation, again we need a citation or the sentence has to start with.. we may speculate...
We added the following, additional reference:
https://doi.org/10.1016/j.amjcard.2015.05.045
- Line 49 – systematic ECG screening in 75 year and older is IIaB recommendation, but opportunistic screening in ≧ 65 years old is IB. The authors may leave this statement as it is, but the level of evidence is quite weak and younger patients should be screened and treated beforehand
We added “…(Class IIA recommendation; level of evidence B)…» on line 52. We agree that opportunistic screening in younger patients is important and is recommended in the current Guidelines. However, as this was not the aim of the present study we chose not to mention it.
- Line 214– study by Tavernier R et al. Heart 2018 Apr;104(7):588-593 should be stated, the prevalence of AF in hospitalized patients was even higher (46%)
We did not mention this study in the initial version of the manuscript because it describes the feasibility of screening with a hand-held device in a geriatric population and not a screening strategy in a defined patient population. In fact, AF was already known in most patients with “screen-detected” AF in this study. A new diagnosis of AF with the handheld device was made in 13% of patients (28 out of 221 patients).
We added the following sentence and reference on lines 245-250:
We are aware of only one study in hospitalized patients, which assessed the feasibility of screening a geriatric population for atrial fibrillation with a hand-held device and reported a prevalence of screen-detected atrial fibrillation of 13%.[doi:10.1136/heartjnl-2017-311981]
We changed the sentence on lines 249-251 as follows: The prevalence of clinical atrial fibrillation in the populations of the above-mentioned studies is markedly lower than in our study: 14.2% in the ARIC study and 9.3% in the STROKESTOP study.
- Add the results of the LOOP trial to the discussion section, and please comment on the treatment of screen detected AF
We added the results of both the STROKESTOP and the LOOP Trial on lines 271-289 as follows:
Recently, the results of two randomized controlled trials have been published, which investigated the benefit of screening for atrial fibrillation with subsequent anticoagulation of screen-positive participants. In the STROKESTOP study mentioned above, screening was only performed at study inclusion for 2 weeks and was positive in 3% of screened patients.[please include the following reference: https://doi.org/10.1016/S0140-6736(21)01637-8] Due to the study design, only half of the participants in the active arm were actually screened for atrial fibrillation. Nevertheless, screening showed a small net benefit compared to standard of care after several years of follow-up. The second trial was the LOOP study, in which an implantable cardiac monitor resulted in screen-positive atrial fibrillation in almost one third of participants in the active arm.[please insert the following reference: https://doi.org/10.1016/S0140-6736(21)01698-6] The primary endpoint of time to first stroke or systemic embolism failed to show a benefit. Importantly, the incidence of atrial fibrillation was 12% in the comparator arm of usual care, and only 6% of patients in the active arm had maximal duration of atrial fibrillation episodes of >24 hours at any time.[please insert the following reference: https://doi.org/10.1016/j.ahj.2019.09.009] In fact, short episodes of atrial fibrillation during continuous monitoring as recorded by an implantable cardiac monitor seem to confer a lower risk compared to longer episodes or higher atrial fibrillation burden. .[please insert the following reference https://doi.org/10.1161/circulationaha.119.041303]
- Do the authors think that all screen-detected AF patients should be treated with anticoagulation? Does it depends on the lenght of the Afib episode or the risk profile of the patients?Because sceening without setting some therapeutic goal does not make sense.
We think that patients with screen-detected atrial fibrillation by a Holter ECG have an atrial fibrillation burden that is high enough to justify oral anticoagulation. This is not the case for screen-detected atrial fibrillation in patients monitored by an implantable cardiac monitor, as patients with very low atrial fibrillation burden and with only short episodes of atrial fibrillation are also detected, in which the benefit of oral anticoagulation is lower. We changed the manuscript as follows on lines 294-299:
If atrial fibrillation is detected by repeat 7-day Holter ECG as in our study, these patients usually have an atrial fibrillation burden that is high enough to justify oral anticoagulation therapy. [please include the following reference: 10.1161/CIRCULATIONAHA.119.044407] A Holter ECG has the additional advantage to provide unequivocal ECG documentation of atrial fibrillation as required by the guidelines.
Round 2
Reviewer 2 Report
no other suggestions, the authors revised the manuscript accordingly